# Barriers in accessing family planning services in Nepal during the COVID-19 pandemic: A qualitative study

Anil Sigdel[1]☯*, Anu Bista[2]☯, Hardik Sapkota[1]☯, Edwin van Teijlingen[3]

1 Department of Public Health, Chitwan Medical College, Bharatpur Chitwan, Nepal, 2 Youth and Comprehensive Sexuality Education, Family Planning Association of Nepal, Lalitpur, Nepal, 3 Centre for Midwifery, Maternal & Perinatal Health, Bournemouth University, Poole, United Kingdom

☯ These authors contributed equally to this work.

* sigdel.aanil@gmail.com

**Data Availability Statement:** All relevant data are within the paper and its Supporting Information files.

**Funding:** The author(s) received no specific funding for this work.

## Abstract

### Background

The COVID-19 virus is still with us, and in resource-limited countries, like Nepal, resurgence of a new variant is still a threat. In this pandemic, low-income countries struggle to provide essential public health services, including family planning. This study was conducted to explore what sorts of barriers are faced by women needing family planning services in Nepal during the pandemic.

### Methods

This qualitative study was conducted in five districts of Nepal. Telephonic in-depth interviews were conducted with 18 women of reproductive age (18–49 years) who were the regular clients of family planning services. Data were coded deductively using the preexisting themes based on a socio-ecological model (e.g., individual, family, community, and health-facility levels).

### Results

Individual level barriers included low self-confidence, inadequate knowledge on COVID-19, myths and misconception related to COVID-19, limited access to FP services low priority to SRH services, low autonomy in family and limited financial ability. Family level barriers comprised of partner's support, social stigma, increased time at home with husbands or parents, un-acceptance of family planning services as essential health services, financial hardship due to loss of jobs, and communication with in-laws. Movement restrictions and transportation hindering access, unsecured feeling, violation of privacy, and obstacles from security personnel were the community level barriers and unavailability of preferred choice of contraception, increased waiting time, limited outreach services by community health workers, limited physical infrastructures, the behavior of health workers, stock out of commodities, and absence of health workers were health facility level barriers.

**Competing interests:** The authors have declared that no competing interests exist.

## Conclusion

This study highlighted key barriers faced by women in seeking family planning services during the COVID-19 lockdown in Nepal. Policymakers and program managers should consider strategies to ensure continued availability of the full method mix during emergency, particularly since disruptions may go unnoticed and strengthen the provision of services through alternative service delivery channels to ensure sustained uptake of such services in this sort of pandemic.

## Introduction

The first case of novel coronavirus (COVID-19) was reported in China in December 2019 [1]. On 11[th] March 2020, the World Health Organization (WHO) declared COVID-19 to be a pandemic and recommended several measures to prevent its transmission [2]. Nepal reported its first case on 13 January 2020 and by 15th November 2021 Nepal has cure rate of 97.7%, and 1.4% case fatality rate [3]. A total of 983,103 has been diagnosed COVID-19 positive in Nepal. The total cases per million population is 32,541; deaths per million population is 396 and test per million population is 191,834 in Nepal [4]. A total of 87,722 positive cases has been recorded in five study districts (Dang, Jhapa, Doti, Baidati and Kathmandu) which holds about 39% of the total cases in the country and 42% of female were tested positive in these five districts [3].

Women's reproductive health is one of the most important aspects of development. The mortality in women of childbearing age can be reduced if they have access to proper maternal and neonatal health services and family planning (FP) [5]. More than 500 women die every day during pregnancy and childbirth in countries due to a lack of trained birth attendants, medical services or unsafe abortions [6].

A recent projection of United Nation Population Fund (UNFPA) forecasts catastrophic impact on women's health if COVID-19 pandemic continues since more than 47 million women could lose access to contraception, leading to 7 million unintended pregnancies [7]. The UNFPA stated that COVID-19 pandemic could critically undermine progress made towards Sexual and Reproductive Health (SRH). In March 2020, an estimated 450 million women are using modern contraceptives across 114 priority low- and middle-income countries. Strategies and social distancing approach adapted by countries to reduce the COVID-19 pandemic transmission is anticipated to affect the ability of these women to continue using contraception [8].

Similarly, there is a greatest unmet demand for SRH services among the most vulnerable—adolescents, poor communities, those living in rural areas and urban slums, people living of Human Immunodeficiency Virus (HIV), internally displaced persons and those experiencing humanitarian disasters [9]. To maintain one's sexual and reproductive health, people need access to accurate information and the safe, effective, affordable and acceptable contraception method of their choice [10]. Problems are further intensified by the existing capacity issues in health care systems [11]. Access to SRH care goes beyond the availability of contraception; education on reproductive health and responsible parenthood, antenatal care, safe delivery and postnatal care, safe abortion, treatment of infertility, treatment of sexually transmitted infections (STIs) [6]. Often the minimum initial service package (MISP) is not being enforced consistently, and the key SRH aspects such as abortion, contraceptives, and supporting young people are being ignored [12]. It was noted that resources of many countries were focused on

the pandemic response and regular services such as SRH services seems were of less priority. Even when SRH services were available barriers to seeking services include movement restrictions/lockdowns and fear of the virus [13]. The COVID-19 pandemic has adverse effects on the supply chain for contraceptive commodities by disrupting its manufacture and by delaying transportation of contraceptive commodities [14]. Also, the closure of health facilities, unavailability of medical staff, and women themselves being hesitant to travel to and visit health facilities due to concerns about COVID-19 exposure will impact women's access to, and continued use of, contraception [8]. A study conducted in four countries namely Nepal, Malawi, Niger, and Uganda found that between 27% and 44% of women surveyed in each country highlighted that their decision to delay or avoid pregnancy has been affected by the Pandemic [15]. Further, analysis of routine health information data reflects that there has been significant reduction in uptake of family planning services in Nepal. The study highlighted that the use of pills has been decreased by 25%, injectables (21%), IUCD (86%) and implant (89%) in Nepal [16]. Hence, this study aims to explore barriers faced by women while seeking family planning services in Nepal during COVID-19 lockdown.

## Materials and methods

A qualitative approach was applied since family planning is a sensitive topic, which can be best captured through careful probing using in-depth interview. Interviews explored the barriers to accessing FP services by women in Nepal during COVID-19 pandemic using telephone interview. Telephonic interview was conducted because of travel restriction imposed by the government to limit the spread of COVID-19. Moreover, considering the geographical terrain where the likelihood of internet connectivity and availability of smart phone among women, we concluded that doing interviews by telephone was the best options for data collection during the study period. Participants were women of reproductive age (18–49 years) who were the regular clients (had continuously used any SRH services for at least six months before lockdown) of the selected SRH clinics. Respondents were selected considering several parameters (education and economic status, caste/ethnicity, accessibility to health services, occupation, methods of FP services and age) that directly have impact on the objective of the study. However, there could be a chance of participant bias (acquiescence bias and social acceptability bias) during data collection. Appropriate measures (pre-testing, expert consultation, maximum use of open-ended questions and limitation of leading questions) were ensured during data collection to minimize those biases.

### Study setting

Nepal announced a national lockdown on 24 March 2020 to late Aril 2021 as a measure to control community transmission of COVID-19, prohibiting domestic and international travels, closure of the border and shut down of non-essential services. During this period, very few clinics offered SRH services to women and therefore, data were collected from different clinics located in five districts across the three geographical areas (Mountain, Hills and Terai) of Nepal. These clinics are the outlet of one NGO working on SRH whose clinics are accredited by Family Welfare Division of Nepal and has stayed open during the lockdown. These clinics from five districts were selected purposively based on the high flow of clients during the lockdown period. Clinic representatives were asked to inform women about the present study. Women consenting to a telephone interview were contacted for appointment. There are more mobile phone connections, about 38.61 million than people, 1.3 connections per person [17]. The recent Demographic and Health Survey highlighted that 72.6% of women have their own mobile/telephone in Nepal [18], whilst the contraceptive prevalence rate is 53% and the

proportion of married women using modern FP is 43% and the unmet need for family planning is 24% [18].

## Data collection

Data was collected by the study team themselves. The study team composed of four members including one female member and a professor. Three out of four team members are Nepali and are public health professionals with more than seven years of experience in SRH field. Semi-structured in-depth interview (IDI) guide was developed in English and then translated into Nepali language. The quality of a translation was verified by a co-author using back translation. Original and back-translated tool was then compared for consistency by the Principal Investigator (PI). The IDI guide was pre-tested with three women seeking FP services in Sunsari using telephone interview and necessary changes were made on the IDI guide based on pre-test findings. Telephone interviews were conducted in Nepali by female co-author and recorded with the women's consent from January 15 to January 25, 2021. Each interview lasted between 30–40 minutes. Telephone interviews were the only option for data collection during the peak of COVID-19. The Study team reflected that the respondents took a bit more time to speak openly compared to face-to-face interviews. Likewise, the pre-consent and advance scheduling (time and date) of interview support in timely completion of interview. In addition, place and timing of interview need to be considered, which will help to avoid the crowd and ensure confidentiality. However, it was also reflected that the facial expressions, which is one of the important aspects of qualitative interview could not be reflected during telephonic interview. To overcome this limitation, follow-up questions need to be carefully asked to get more insights during telephonic interview compared to face-to-face interviews. Saturation was reached with 18 participants (see Table 1). We noticed that the information was saturated because the last three interviews did not add any new information complementing the objectives and scope of the study. The interview guide was divided into four sections (individual level barriers, family level barriers, community lever barriers, organizational level barriers), based on the concept of socio-ecological model for exploring barriers to FP services as used in previous studies as shown in Fig 1 [19, 20]. Once the data collection from each district were completed, we share some of the key findings with health services providers to validate the findings. Furthermore, overarching lesson learned from the study was also reflected in the result section.

## Data analysis

The recordings were transcribed verbatim in Nepali language by co-author (female) who conducted interviews, which were then cross-checked by PI and one team member and translated into English and thematically analyzed by two authors using a deductive approach [21], using the socio-ecological model (e.g., individual-, family-, community- and, health facility levels), see **Table 2**. Transcripts were then re-read by two co-authors to identify and discuss emerging themes among the research team, to increase validity. Thematic coding [21] was based on the key concept of barriers in accessing to FP services; similar codes were categorized and clustered in sub-themes as in **Table 2**. These formed the initial coding frame, broadly related to barriers in accessing FP services that is from individual level to organization level.

## Ethical considerations

Ethical approval for this study was granted by Nobel College, Kathmandu, Nepal. During visits to health facilities, each interviewee was given a one-page information sheet, which described the purpose and process of the study. The clinic representatives explained the respondents

**Table 1. General information about study participants.**

| ID | Region | Age | Ethnicity | Education level* | Occupation | Economic status** | Number of Children | Types of Client | Types of FP Use |
|----|--------|-----|-----------|------------------|------------|-------------------|--------------------|-----------------|-----------------|
| 1 | Terai (Dang) | 24 | Janajati | Higher Secondary | Housewife | Medium | 1 | New | Short-term |
| 2 | Terai (Dang) | 30 | Brahmin/ Chhetri | Bachelor | Employed | Good | 2 | Current | LARC |
| 3 | Terai (Jhapa) | 33 | Brahmin/ Chhetri | Higher secondary | Grocery shop owner | Good | 2 | Current | LARC |
| 4 | Terai (Jhapa) | 18 | Janajati | Secondary | Daily Wages | Poor | 0 | New | Short-term |
| 5 | Hilly (Doti) | 27 | Dalit | Literate only | Daily Wages | Poor | 3 | New | LARC |
| 6 | Hilly (Doti) | 39 | Brahmin/ Chhetri | Literate Only | Agriculture | Medium | 3 | Current | Short-term |
| 7 | Baitadi (Hilly) | 21 | Brahmin/ Chhetri | Higher Secondary | Agriculture | Medium | 1 | Current | Short-term |
| 8 | Baitadi (Hilly) | 25 | Dalit | Primary | Daily Wages | Poor | 3 | New | Short-term |
| 9 | Kathmandu Valley | 35 | Adabasi (Newar) | Master | Business | Good | 2 | Current | Short-term |
| 10 | Kathmandu Valley | 29 | Brahmin/ Chhetri | Primary School | Daily wages | Poor | 4 | New | Short-term |
| 11 | Kathmandu Valley | 37 | Dalit | Secondary | Housewife | Medium | 2 | Current | Short-term |
| 12 | Kathmandu Valley | 19 | Madheshi | Higher secondary | Housewife | Medium | 0 | New | Short-term |
| 13 | Kathmandu Valley | 28 | Adabasi | Bachelor | Employed | Good | 2 | Current | LARC |
| 14 | Dang (Terai) | 24 | Madheshi | Primary | Daily Wages | Poor | 2 | New | Short-term |
| 15 | Dang (Terai) | 34 | Janajati | Secondary | Housewife | Medium | 2 | Current | LARC |
| 16 | Doti (Hilly) | 23 | Dalit | Secondary | Agriculture | Poor | 3 | New | Short-term |
| 17 | Baitadi (Hilly) | 32 | Brahmin/ Chhetri | Primary | Housewife | Medium | 3 | New | LARC |
| 18 | Jhapa (Terai) | 24 | Madheshi | Secondary | Agriculture | Medium | 2 | New | Short-term |

*those who did not attend formal education but can read and write are considered 'literate only respondent', education up to grade 5 are considered as primary, education up to grade 10 are called secondary, education up to grade 12 is 'higher secondary', undergraduate is 'Bachelor' and the highest level was Master.

**self-reported by respondents in the study.

about the consent and confidentiality, that the participation was voluntary and anonymous and that they had the right to refuse any question or end the interview without further explanations. The clinic representative will then obtain the written consent with the participant. To ensure the confidentiality, the names were replaced with a code in all recording, all the personal identifiers were removed from the records and records were deleted soon after the completion of translation.

## Results

Barriers in accessing FP are presented at four different levels as outlined in **Table 2**.

### Individual level barriers

The individual level barriers included low self-confidence in seeking FP services, insufficient knowledge of COVID-19, low priority to FP services, lack of autonomy/decision making and money.

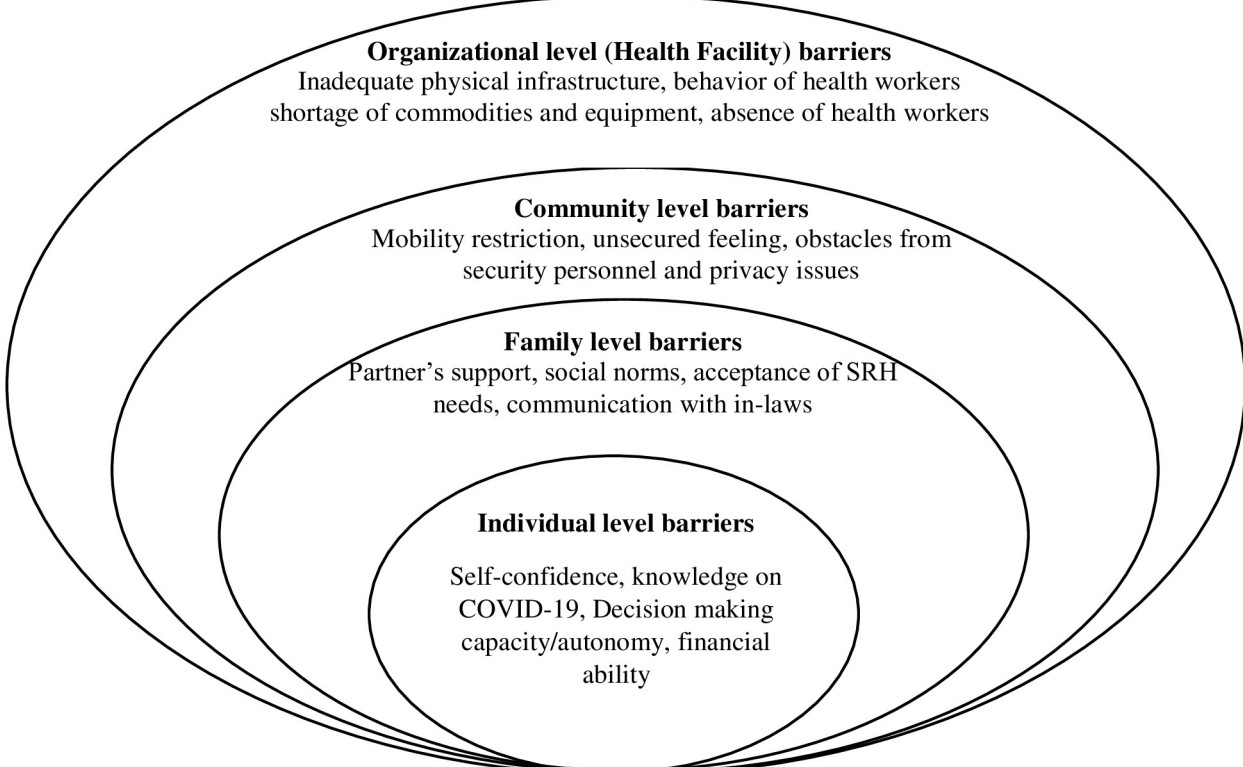

**Fig 1. Socio-ecological barriers in accessing SRH services adopted from USAID and health community capacity collaborative.**

### Low self-confidence on seeking SRH services

Participants were in a state of uncertainty as to whether or not to seek the FP services. Their primary reasons for low self-confidence were wrong information such as people will die if they have an infection, there is no any cure for the infection, is asymptomatic in transmission, can transmit very quickly from different means, and prevailing COVID-19-related stigma and discrimination. Often, assuming that the health facilities were the first point of contact for patients with COVID-19 (asymptotic / symptomatic), this prevented respondents from deciding to pursue FP services.

> *Initially, I thought that it is safer to stay at home without using any FP measures rather than increasing exposure to COVID-19 by visiting health facility. However, later realized that the long-term impact of not using contraceptive measure will be more risky than COVID-19 and convinced myself for the services before talking with any family members including partner. [Participant (P.)15].*

### Limited access for support

The provision of lockdown and restriction of movement restricts participant's access to support, for instance they were not able to meet friends and share their problems related to family planning. Further, participants also highlighted that there were very limited messages/information related to SRH during pandemic, most were related to COVID-19. In addition, the participants mentioned that health workers were one of the mostly affected groups and it is not safe to visit them to seeking such services.

**Table 2. Coding tree.**

| Theme | Sub-theme | Codes |
|---|---|---|
| **Individual level barriers** | myths and misconception related to COVID-19 | Misinformation on COVID-19 regarding Risk of death; lack of treatment; is asymptomatic in transmission; can transmit very quickly from different means; COVID-19 stigma and discrimination. |
| | Limited access for support | Unable to meet friends and share problems, fears, and stigma of meeting with health workers, limited messages on SRH services |
| | Myths and misconception related to COVID-19 | COVID-19 transmission, fatality, prevention and mitigation measures vary in media; diverse messages make people more phobic and restrict their movement to seek care |
| | Low priority to SRH services | Delay in up take FP services has less immediate impacts than injuries; Going to facilities increases exposure and chances of COVID-19 transmission; Seek FP after things being normal |
| | Low autonomy in family | Position of women in family impact on pursuing FP services; Higher educated and employed were more comfortable in making decision on seeking services; Structure of family (affects decision-making) |
| | Financial hardship | Poverty effects in pursuing FP; Daily wage earners; wives of migrants are less likely to uptake FP; basic needs are priority than spending money on contraceptives. |
| **Family level barriers** | Support from Spouse | Need to get permission from husband to seek FP services/ go outside; response to family on reasons behind visiting HFs; Partner's positive approach supports in seeking services |
| | Social Stigma | Prevailing social norms that anyone going to HFs will carry COVID-19 home and spread the disease; FP measures have health side-effects /lead to infertility; society will know that you use FP this will hamper social status. |
| | Un-acceptance of FP as essential by family | You won't die without using FP; you may die because of COVID-19 if you go and infected; having children is better than having COVID-19 to all family members |
| | Increased time at home with husband and in-laws | Take care of regular household chores and husband/parents limiting time for service uptake |
| | Communication with in-laws | Cannot communicate openly with in-laws; talking about FP is against social norms, especially women (daughter in laws) speaking to senior family members |
| **Community level barriers** | Movement restriction and transportation hindering access | Lockdown halts movement of vehicle; long travel times for seeking services |
| | Unsecured feeling (fear of transmission from other) | Others not using masks; feel at risk of infection; people not maintaining physical distancing |
| | Obstacles from security personnel | Needs to answer frequently to security personnel regarding their movement; sometimes verbal assault from security personnel passing unnecessary comments |
| | Confidentiality Issues | Movement during lockdown; several rounds of questions and answers with security personnel; interaction with people in the way void the confidentiality in seeking services. |
| **Health facility level barriers** | Physical infrastructures at health facilities | Not adequate space for maintaining the physical distancing while waiting for services; facilities crowded in the day of immunization; limited facilities with hand washing facilities. |
| | Increased waiting time | Physical distancing measures, provision of checking up limited numbers of clients per day |
| | Behaviour health workers | Very limited counseling on FP provided; HWs mostly focus on COVID-19; request the clients to report to concern authorities if someone is suspected of COVID-19 |
| | stock out of commodities/supplies | Unavailability of commodities and equipment in facilities especially long acting reversable contraception; unavailability of preferred choices of contraception |
| | Absence of health workers | Unavailability of trained staff in health facilities; staff stuck at home in lockdown or assigned to COVID-19 hospitals and quarantine sites, limited outreach services by community health workers. |

*I am a newly married and currently initiated using short-term family planning methods. However, I experienced some difficulties after its use. I have limited information regarding side effects and I could not meet with my friends to discuss on it. This result in discontinuation of method for few months. [P. 12].*

## Myths and misconceptions related to COVID-19

There found that the havoc messages (messages that creates unnecessary threats) related to COVID-19 hinder women's decision in seeking FP services during COVID-19 pandemic lock-down period in Nepal. Respondents told that the messages on COVID-19 transmission,

fatality, prevention and mitigation measures varies in different medias. These sorts of diverse messaging make people more phobic and restrict people's movement for seeking care. But contradictory to other respondents, one woman told that different social medias help in message dissemination and alert people on COVID-19 that eventually supported in seeking FP services from nearby health facilities.

*The information on transmission, fatality and stigma related to COVID-19 on different social medias like Facebook, YouTube makes me to decide on not seeking FP services during this pandemic. [P. 18].*

## Perceived importance to SRH services

The information from respondents indicated that the women perceived FP services as less important than any other health concerns because it does not have an immediate direct impact on health and illness. In this view, a few women explained that it is better to say at home and reduce the risk of transmission of COVID-19 than increasing risk to COVID-19 by going to the health facility for FP services. They also indicated that it is better to seek FP services after things would get back to normal rather than seeking services during COVID-19 peak time.

*I think that it's not a good idea to seek the FP services during this lock-down period as this does not have any immediate impacts like injuries, burns, etc. You can choose other traditional methods and when things come to normal and the lock-down ends, we can go for the modern contraceptives. [P. 3].*

## Low autonomy in the family

Respondents told that the position of women in the family explicitly or implicitly prohibits them from using FP services. Respondents told that highly educated and employed women were more comfortable in making decision in utilizing FP services than those who were less educated and housewives. In addition, the respondents indicated that the structure of family have affected to pursue FP services; women in joint family were less likely to make their decision for using SRH services than the women in nuclear family. However, some of the respondents told that, it is not the structure of the family but the decision to seek FP services depends on oneself.

*One of my nearby friends who is a housewife says that she wishes to up-take the family planning measures from the nearest clinic but cannot discuss to her family members because they avoid any outside movement due to fear of COVID-19 transmission. She also told that the family members never consult her while making any decisions and she needs to get permission from her in-laws while going outside the home. [P.17].*

## Financial hardship

According to respondents, poor women were less likely to pursue FP services than women with decent financial position. They told that women who are working on daily wages, wives of migrants, are less likely to go and seek FP services during lock-down because their priority during this lock-down is fulfilling their basic needs for the family rather spending money on contraceptives.

*I lived in a community where male members from majority of the household go to India as seasonal migrant for living. One of my friends whose husband is also a labor migrant had recently returned home during lockdown. To have safe sexual life, she expressed her wish to have FP services but unfortunately could not go for services. When I asked her the reason for not going to clinic for FP service despite of having interest, she told me that she had to do calculated expenses for basic needs, as there is uncertainty in opening lockdown. Despite the service is free, women like her cannot afford transportation cost to reach to the clinic. [P.8].*

## Family-level barriers

Family life and marriage are typically dominated by patriarchal norms, laws, traditions, and gender differences. Women in Nepal have lower social status than men, which has a negative effect on a woman's ability to claim rights and make decisions regarding their sexual and reproductive health decisions [22].

Partner's support, social norms, family acceptance of FP needs, and the communication with in-laws were the barriers for seeking FP services during the lock-down in Nepal.

## Support from the spouse

All the respondents interviewed were married and they indicated that they need to inform and get permission from their spouse to pursue FP services even during the normal condition. Women told that they need to ask their spouse to go outside mostly in this pandemic. They had told that it is not possible to seek services without the partner's permission during this lockdown as every member of family will ask you the reason behind going to health facilities. Partner's positive approach makes it easier for women to access and use FP, and as a result, availability and continuity in services is ensured.

*I told my husband that I must go to the health facility for up-taking depo (Injectables) since it has already been three months that I have injected and needs to continue. My husband was quite offensive initially arguing that the FP is not the current priority when everyone is struggling with COVID-19 situation. But when I told him that some of the nearby health facilities are offering the FP services with necessary precautions and continuation of Depo will help in our sexual and reproductive health, he was then convinced and we both went to the health facility for the FP service [P.12].*

## Social stigma

Many anthropologists argued that reproductive decisions made in relation to family planning choices are not only dictated by economic factors, but also influenced by sociocultural factors such as desire for reproduction or beliefs related to having children. Anthropologists subsequently, stressed that it is very important to consider what social, cultural or structural factors may influence the people's thoughts and behaviours [23].

Social norms are informal, often implicit rules that most people accept and abide by [24]. Respondents told that the prevailing social norms and society thinking that anyone going for the FP services during this lock-down, will carry COVID-19 at home and spread the disease. In addition, there is a belief that one should not use FP measures before the birth of first children as FP will leads to infertility, thinking FP measures have health side effects. These beliefs are influencing the women in seeking FP services.

*One day I was ready to go to health facility for FP service. Meanwhile, my mothers-in-law asked me where and why I was going. I replied her that I was going to health facility for FP service, then she told me not to go during this period because everyone will know that you are using FP measures even when the people are thinking about how to get out of this COVID-19 emergency, and this will hamper our social status. [P.6].*

*One of my newly married friends told me that she was not allowed to go to health facility for seeking FP services. When she talked with her husband for up-taking FP services, he told that she needs to take permission from mothers-in-law. When she asked her mothers-in-law, she told that newly married women with no child should not use any FP measures because this could lead to infertility. [P.1].*

## Un-acceptance of family planning as essential health services by family

Respondents in the study told that family members initially do not allow us for going outside and seek FP services stating that you will not die without family planning services. However, you may die because of COVID-19 if you go and get infected. Even, the family members from literate household does not consider FP as essential health services and encourage women in seeking services. They told that having children is better than having COVID-19 to all family members.

*One of my friends was not allowed to seek the FP services by the family members telling her that the risk of transmission of COVID-19 is high if she goes to health facilities. It's better to have child then to have COVID-19 to all family members.* [P.3].

*I belong to the economically strong family, however when I told my husband that I need to go to health facility for FP services, he opposed me for seeking services. He said that it's better to have one more child whom we can afford than COVID-19. [P.10].*

## Increased time at home with husband and parents

Respondents highlighted that they were more occupied with household chores and other agricultural work during the pandemic context. Further, they also highlighted that they need to spend significant proportion of their time taking care of their in-laws and husband in addition to their children. This additional task limit the available neither time to seek services nor could they easily go outside the house without their consent, which restricted the uptake of services.

*I can go to health facility to seek services as my in-laws and husband go for work before pandemic. The movement restriction and lockdown made them stay full time in the house which makes me difficult for moving outside the home and even for seeking FP services. [P. 12]*

## Communication with in-laws

Respondents in this study believed that sometimes the relationship and communication gaps between daughters-in-law and in-laws often existed in the joint family and in the household where the in-laws are of old age with high traditional beliefs. Respondents told that they cannot communicate openly with their in-laws regarding FP services as it being considered as personal and it is against the social norms. In addition, women were considered shameless when they talked about the FP measures with senior members of the family.

*My mothers-in-law is well educated and a retired government employee. She treats me like a daughter and frequently asked me regarding our plan on having child. She encouraged me to have planned baby and supported me in using FP services. However, this is not the case with most of women in our community. Women were considered shameless when they talk about the FP services openly in community and in home. [P.3].*

*One of my friends was planning to come to health facility to seek FP services with me, but she denied coming to the health facility at the last moment. When I asked her the reason, she told that she could not speak to her mothers-in-law to get permission for seeking services. [P.6].*

## Community-level barriers

Respondents told that they encountered several community level barriers while seeking services during the lock-down. These are the most prominent barriers that hinders the women in seeking FP services. Movement restriction during lock-down, unsecured feelings, obstacles from security personnel and the privacy issues were the community level barriers identified by women in the study.

## Movement restriction and transportation hindering access

The lock-down imposed by the Government of Nepal halted the movement of vehicles preventing women from accessing FP services. Almost all respondents told that the long travel time was the most common problem they encountered while seeking FP services.

*Very limited health facilities are in operation during this lock-down period and to those who are giving services have limited health care services to the emergency care only and the public vehicles were not allowed to operate. We need to walk for more than usual time to go to health facilities and seek services. This is the one of the major obstacles we faced during lock-down. [P.16].*

## Fear of transmission

During their travel, respondents feared they might unintendedly come across to pedestrians who are not wearing masks in public areas. Meeting such people during our travel, made them feel unsecured and worried about possible transmission. Women indicated that the movement of people without using minimum preventive measures, e.g. not even maintaining physical distancing, stopped them from seeking out services.

*I was going to the health facilities for seeking FP services. During my way to health facility, I saw people not using any protective measures (even mask). Every time I come across them; I feel that I might get infected with COVID-19; which is said to be highly infectious and make me feel vulnerable and unsecured. [P.13].*

## Obstacles from security personnel

Security personnel policed the lock-down. Respondents told that they need to answer to army or police personnel at many points (several times) regarding their movement. They too indicated that women were sometimes verbally assaulted by security staff commenting that using FP services was unnecessary.

*On my way to health facility from home, which almost took 45 minutes, I was asked by the security personnel three times on my way about my movement. When I told them that I was going for seeking FP services; one of them commented as "Can't you control your sexual desire till lock-down". I really feel ashamed listening to him. [P. 18].*

### Privacy

Respondents from hilly districts such as Doti and Baitadi told that FP services in their community were considered personal and attendance needs to hidden from the community. They indicated that their movement during lock-down; several rounds of questions and answers with security personnel in the way to service centers and interaction with other people in the way void their right to seek confidential FP services. They told that this might increase the discrimination in the society.

*When I go for services, most of the community people can notice me due to very few movements in the road. When they saw me going for services; they ask me many questions as where and why are you going, what is the urgency to visit the health facility at this emergency time, did you told your husband about your movement, and many more that affect my decision to go health facility. Also, I need to answer the security personnel several times. I think I cannot enjoy my right of seeking confidential FP services in such cases. [P.16].*

### Health-facility-level barriers

Women need to overcome different levels of barriers before reaching to the health facilities for up-taking FP services. Respondents reported that they encountered some barriers existing at the facility level during this pandemic period. Most SRH clinics adopted an on-call approach to reduce the possible exposure to COVID-19. Respondents in this study indicated the inadequate physical infrastructure, behaviour of health workers, stock out of commodities and equipment, and absence of health workers were some of the barriers that they encountered in the health facilities while up-taking FP services.

### Physical infrastructures

Almost all the respondents told that there is not adequate space for maintaining the physical distancing while waiting for services. Although the health facilities have initiated to make handwashing corner and prepare for the physical distancing but during the national priority programmes like immunization day it was hard to properly manage these systems. Health facilities have used infrared thermometer to measure the thermal temperature as another protective measures. The health facilities were much crowded in the day of immunization and has been difficult to maintain physical distancing due to lack of open space. Respondents told that when they saw such crowd they were not motivated for services and did not recommend others to seek services.

Women also noticed that there were not continue operation of hand washing facilities in the health facilities due to large number of people coming for immunization and FP services. Women told that some of the health facilities have tap water but no soap for washing hands and other health facilities, may have a tap but there is no water. Women told that very limited health facilities need to manage WASH facilities for many service users.

*I was in the health facility for FP services on the day of immunization. There were more than 50 people with their children in the health facility who came for immunization services. The waiting space at health facility was small so it was impossible to maintain the physical distancing. I really feel unsecured being in crowd in the health facility. [P.12].*

*I went to wash my hands at the health facility and found that there was no soap in the hand-washing corner and since everyone was busy to provide the health services no one's attention was drawn to pre-positioning of soap. [P.4].*

*I searched for the hand-washing corner in the health facility but did not find any such place. When I asked the support staff of the health facility, he told me that the corner has been closed because of no water supply. [P.6].*

### Increased waiting time

Respondents in this study highlighted that one of the barriers in seeking FP services is long waiting times in health facility because of provision of physical distancing measures and also some of the health facilities limits the number of clients per day to restrict the transmission of COVID-19. This leads to increased time to receive services.

*I have to wait for more than three hours to seek services from health facility during my last visit, which is more than double time compared to normal context. The provision of physical distancing and other protocol delays in delivering of services. I noticed many other clients leave health facility without seeking services. [P. 17].*

### Behaviour of health workers

Women in the study indicated that they did not notice any differences in the attitudes and behaviours of health workers providing services during this pandemic compared to normal days. The only differences they noticed during up-taking services in this lock-down was very limited counseling on FP measures was provided. They noticed that health workers mostly focused on counselling about the COVID-19 than the FP measures. All women in this study noted that health workers emphasized COVID-19, its sign and symptoms and the preventive measures.

*I do not notice any differences in delivering services than usual days. Only the difference I felt was that the health workers were using the personal protective equipment (apron, goggles, mask and gloves) that they usually did not use in normal days before. [P.17].*

*The good thing I noticed during my visit to health facility was health service providers were providing extensive counseling on COVID-19, its sign and symptoms and the preventive measures. They were also requesting the clients to report to concern authorities if someone in the community were suspected for COVID-19. [P.18].*

### Stock out of commodities and supplies

The impact of COVID-19 pandemic have affected the supply chain for the contraceptive commodities by disrupting the manufacture of key pharmaceutical components of contraceptive methods and by delaying transportation of contraceptive commodities [14]. Even the commodities are at the regional and national warehouse, some delay in sending the commodities

was noticed due to mobility restriction and closure of the domestic airport. Respondents in this study told that they cannot receive the desired/preferred FP methods because of unavailability of commodities and equipment in the health facilities. These sorts of cases were found common with female planning for up taking the long acting reversable contraception.

*I decided with my husband to use long acting reversable FP methods (IUCD) and went to health facility to seek services. However, I was unable to receive the desired services because of unavailability of required commodity i.e. IUCD set for insertion. We then, decided to use other short-term FP methods. [P.10].*

### Absence of trained health workers

Respondents told that they cannot get the desired FP services because of unavailability of trained health personnel in the health facilities. They told that many health workers were stuck in their hometown and cannot returned to the service center because of lock-down in the country and some health workers were assigned at the newly formed COVID-19 hospitals and quarantine sites. The women from hilly districts mostly raised these sorts of problems.

*My husband just returned from India during the lock-down, it has been one year he was in India. Since we already have two children and do not want to conceive the third one, we decided to use Implant as FP method. We have visited the nearby health center for the insertion of Implant but unfortunately, we were told that the trained health personnel have been appointed at the quarantine site and there is no one except her. Since it needs to be done by the trained health worker, we were suggested to use either condom or pills for the time being. [P.7].*

### Overarching lesson learnt

The study found that along with the prevailing barriers in the uptake of FP, the global pandemic like COVID-19 have significant role in introducing/adding new additional barriers in uptake of FP services. The emergence of COVID-19 leads to movement restriction, financial hardship due to loss of job, un-acceptance of FP as essential health services, increased time at home with husband or parents, violation of privacy, increased waiting time at health facilities, limited outreach services by community health workers, delayed prepositioning of FP commodity and diversion of health workers to respond COVID-19 were noted as the additional barriers in uptake of FP services by women in low-and-middle-income countries like Nepal. This clearly implies that public health efforts, particularly SRH programs, should consider the additional challenges that occur as a result of the pandemic during the design and execution of public health programs.

## Discussion

Global awareness of and responsiveness to women's sexual and reproductive health and rights during a conflict or crisis were largely lacking; Instead, humanitarian responses made access to food, water, shelter, sanitation and immediate medical assistance [25]. During emergencies, women and girls are at particular risk when social and structural support system around collapses including health system, which consequently increase threats to women's sexual and reproductive health; specifically, expose women and adolescent girls to unwanted pregnancy, unsafe abortion, STIs including HIV and maternal morbidity and mortality [25]. Considering

the high vulnerability of women and adolescent girls to increased risk of poor sexual and reproductive health, current study used socio-ecological model to analyze the possible barriers restricting women to access SRH services. This study acclaimed that individual level barriers such as myths and misconception related to COVID-19, inadequate knowledge on COVID-19 and lack of confidence on seeking services during COVID-19 pandemic were the key barriers of responded women hindering in seeking FP services. The study also reflects that the women belonging to poor family and less educated were mainly affected by the lockdown in seeking services. This result has been supported by the study conducted in Bangladesh, which highlights 23% decrease in prevalence of FP compared to before pandemic data where women with secondary level education have 1.65 times higher odds of using contraception compared to their counter parts [26]. Further, a technical brief on impact of the COVID-19 pandemic on demand for family planning services in Bangladesh indicates that fear and stigma related to COVID-19 are some of the barriers in uptake of FP services in low and middle countries like Bangladesh [27]. Likewise, a scoping study on impact of COVID-19 in low-and-middle-income countries also highlighted those myths and misinformation about COVID-19 play a significant role in decreased uptake of FP services [28]. This clearly reflects that there should be a proper mechanism for dissemination of the right message to the right audience using the right methods to improve the uptake of FP services [29].

Family planning a priority of the Government of Nepal has been enlisted as the part of essential health services [30] which means that FP services will be provided by all governmental HFs with free of cost. Although the FP services were provided free of charge from the health facilities, the study noticed that cost for reaching services and medical cost for IUCD and Implant were one of the barriers among those women whose living is dependent on daily pays and were the wives of labor migrant whose income has been stopped due to lockdown. Contradictory to this finding, the study in Lao in exploring the perceived barriers in accessing SRH services shows that cost related to reaching and using the SRH services were not taken as a major barrier among youths [29]. The contradictory findings might be because of differences in the respondents enrolled in the study i.e. the study in the Lao includes mostly the youths aged 15–24 years whereas the present study includes the women of reproductive age i.e. 18–49 years. However, this result complements the findings from Bangladesh where there is higher proportion (69%) decline for long-acting FP method in the initial months of COVID-19 with annual decline of 31% [27]. Likewise, the study conducted in Guinea on impact of Ebola on FP also found a 51% decline in facility-based contraceptive visits mostly for long acting FP methods during the Ebola outbreak [30]. This reflects that long acting FP methods, which demand more technical skills, were found to be mostly affected during emergencies context.

Another major factor affecting the utilization of family planning services by women during any crisis situation is the status of women in the society and within her family. Women in Nepal reported lower social standing compared to men, which negatively affects a women's ability to claim rights and to make decisions regarding her sexual and reproductive health [31]. This study also indicates that the position of women within her family affects in making decision in seeking FP services. Women from nuclear family, having good education and employed were more likely to make decision oneself than women living in joint family. This result aligns with the findings from a study conducted in Bangladesh where the women with secondary level education had a 1.65 times higher chances of uptake of FP method. Likewise, there is 1.5 times higher chances of uptake of FP services among employed female compared to their counterparts in Bangladesh [26]. This reflects that female empowerment play a significant role in uptake of services even in difficult situations and support in investment on girls' education and empowerment.

The support and concern of a partner during pregnancy can positively impact a women's pregnancy [32]. In Nepal, the husbands and elders in the family makes most of the decisions. These decisions may reflect the interests of the male members of the family rather than interest of women [31]. Current study also shows that the positive partners' support and decision are essential even for women in seeking FP services in normal days and during this lockdown, their roles is critical for women in up-taking services which aligned with the study conducted in Pakistan after 2005 earthquake where pregnant women, who were widowed and who did not have a male relative, found difficult to access antenatal care, delivery care and post-natal care [33]. Husband/partners support was found essential for women in deciding and seeking FP services during normal days and this has been enhanced during lockdown in Nepal. This might be because of the patriarchal norms in Nepal where male was considered as the head of household and considered one to make decision for whole family.

Another longitudinal study in Jamaica found that forty-eight percent of women experienced side effects for one year's follow up which highlights that the occurrence of side effects have negative impacts in the continuation of family planning services [34]. Similar finding was found in a qualitative study of Nepal, which found that the fear of health hazards and side effects of modern contraception was a major problem in up-taking modern FP methods [35]. Similarly, according to the study conducted to explore the influence of side effects and social norms in use of family planning, fear of side effects and misconceptions over FP method side effects were two major barriers for FP uptake reported in in Nepal [35]. This study also indicates that the misconception regarding FP like; one should not use FP until having first child; FP leads to infertility etc. which is another factor holding back women in seeking FP services.

Government of Nepal announced the complete lockdown from March 24, 2020 restricting the movement of people. The restriction of movement and the deployment of the security personnel were found to interrupt access to sexual and reproductive health services [36]. Women in this study told that they need to walk several kilometers for seeking FP services because of restriction of vehicle movement. This finding aligned with the study conducted in Lao where the respondent told that distance to reach service site is the major geographical barriers for accessing SRH services [29]. Likewise, studies on Senegal, Bangladesh, Malawi, Niger, Nepal and Uganda also highlight the impact of restriction of movement and transportations barriers as one of the reasons for low uptake of FP services during COVID-19 period [15, 26, 30]. Women told that they feel unsecured in seeking FP services when they unintendedly come across the pedestrians who are not using any precautionary measures like wearing mask while walking outside, leading women believing that they can get infected. This result resembles with studies, which showed that surgical masks and N95 respirators were similarly effective in preventing influenza-like illness and laboratory-confirmed influenza [37, 38].

Training of providers and adequately equipped facilities before an emergency are necessary so that they can response during the outbreak of any crisis and with the fact that inadequate commodity management systems may be unable to deliver reproductive health supplies [29]. Several studies also highlighted that lack of supplies are one of the prominent factors for low uptake of FP services mostly during the emergency period [15, 28, 30].

Stock-outs of preferred contraceptive methods and unavailability of long-acting reversible contraceptives (LARCs) in some facilities, negatively affected contraceptive utilization, as it meant that communities could not use nor access such services when they wanted to [39]. This result aligned with the finding from the current study where women noticed that they compromised their choices because of unavailability of FP commodities and equipment in the health facilities. Likewise, women told that the unavailability of trained health workers hinders in utilization of services, which aligns with the study conducted in Zambia that reflects the

availability of trained health workers were perceived to be an enabler, as it provided a basis for provision of the minimum method mix to the community members [39].

Nepal Health Facility survey reveals that undesirable health care worker's attitude stands as a barrier to FP/C services utilization, especially for marginalized user groups, like the unmarried and adolescent users. The health workers and key community stakeholders reported that negative attitudes such as shouting, scolding, not allowing clients to explain their side effect experiences, and giving preference to socially accepted FP/C services user groups like the married women, existed in some of the health facilities [40]. The present finding also shows that the health workers mostly focused on counseling on COVID-19 rather than providing information on side effects and different means of family planning. Shifting of the priority has affected the use of FP services. This finding reassembles with the health facility survey conducted in Nepal, which reflects that method-specific side effects were discussed in a little more than one in five FP consultations [40].

Health facility survey reflects that hand-washing supplies were seen in just over half of health facilities offering FP services in Nepal [40]. Women in this study believed that the inadequate space for physical distancing and limited availability of hand washing facilities makes them feel unsecured in seeking FP services during COVID-19 lockdown period in Nepal. The integration of National Immunization programme in SRH clinics has also compounded the availability of hand washing commodities and physical distancing.

FP being a sensitive issue in Nepal, several studies have highlighted that the service provider from opposite sex can influence the uptake of FP services [41, 42]. Use of female co-author who understand all three local languages used by respondents while collecting data through telephonic interview in pre-determined time minimized the language biases and other potential biases likely to occur due sex of researcher. Also, we noticed that the rapport building in their local dialect helped to build trust among the respondents and improve the data quality. Further, verifying sample interview recordings and transcripts by other authors was helpful to ensure quality of data during analysis. In addition, validating the key findings from the study with the clinical representatives of the health facilities support to refine the findings and codes. Moreover, prior ethical approval from the respondents and interviewing respondents in their proposed time were really helpful to collect the data even with virtual mode.

One of the predominant reasons for low uptake of FP services during pandemic in Nepal is that FP services are not considered prioritized or essential services either by individuals, or by families. This finding resembles the study conducted in Bangladesh on the impact of COVID-19 which also highlights that FP was not considered a priority while comparing with other maternal health services like Antenatal or postnatal care [27]. The increase in waiting time has also been shown to be one of the important reasons for low uptake of FP services in Nepal which has been supported by a scoping study conducted in low- and middle-income countries [28]. In addition, this study highlighted that limited outreach by community health workers resulted in limited accessibility of FP services which too has been supported by a scoping study conducted in low- and middle-income countries [28]. This reflects that uptake of FP services in affected by different dimensions, even more critically during emergency contexts.

## Limitation of study

There are several limitations of this study. First, though this study provides insights in exploring the barriers in uptake of FP among the sample of women living in low- and middle-income country during COVID-19 with particular focus in Nepal, who could participate in telephone interview, its qualitative findings were not designed to be generalized. Considering the context of COVID-19, movement restriction and sensitivity of the topic, participants may have been

more reluctant to share information that they could have shared more openly in face-to-face interview. However, our study was a cross-section study, we did not have an opportunity to ask follow-up questions and questions focused on asking participants to share changes during and after COVID-19. Although we explore the different experiences of married women, we could not draw experiences of unmarried adolescent/women during the pandemic, which is an important area for future research.

## Conclusions

This study documented the challenges women encountered on seeking FP services during the COVID-19 pandemic in Nepal. The major individual-level barriers faced by women were: low confidence in seeking FP services, inadequate knowledge on COVID-19, low priority of SRH services, limited financial ability and low autonomy. Partners' support, prevailing social norms, acceptance of FP services as essential services by family members and communication with in-laws were key family-level barriers women faced in seeking services. Whereas, community-level barriers included movement restrictions, unsecured feelings, obstacles from security personnel and privacy. Limited physical infrastructure, behaviour of health workers, stock out of commodities and equipment and absence of health workers were facility-level barriers in seeking FP services. In order to better understand the role of partners, family, and friends, as well as how this influences changes during the time of crisis, stress, and uncertainty, SRH programmes should commit to interacting with societal and gender norms that shape FP access. The role of interventions in changing social norms held by partners and parents needs to be considered and peer-to-peer opportunities invention can be explored. Social media and online platforms are still mostly untapped but offer ways for women to get FP knowledge. Furthermore, given different levels challenges (interpersonal, financial and structural challenges) that women face in negotiating access to FP, FP should be considered as essential services in health facilities. In the event of COVID-19 or subsequent pandemics, government health facilities, non-profits, and other SRH actors and donors should make a commitment to improving the effectiveness of FP delivery systems in order to avoid delivery interruptions and reduce associated costs. This will be crucial to ensure that the progress made in sexual and reproductive health are not reversed. This study emphasized the needs to incorporate SRH into multi-sectoral and health disaster planning and implement SRH services considering the impact of COVID-19 lockdown, to ensure sustained uptake and meet the populations' needs. Future studies should focus on how COVID-19 has increased disparities in access to FP in different sub-groups of population, explore the unmet need of FP as result of financial stress and uncertainty caused by COVID-19, and identify the changes to social and gender norms related to FP decision making.

## Supporting information

**S1 File.**
(PDF)

## Acknowledgments

The authors would like to thank Basanta Khanal, Chetraj Phulara, Uddav Dev Bhatta and Sarad Aryal for their support in collecting data. Further, we would like to extend our gratitude to all the respondents for their time and cooperation.

## Author Contributions

**Conceptualization:** Anil Sigdel, Anu Bista.

**Formal analysis:** Anil Sigdel, Anu Bista, Hardik Sapkota, Edwin van Teijlingen.

**Investigation:** Anil Sigdel, Anu Bista.

**Methodology:** Anil Sigdel, Hardik Sapkota.

**Writing – original draft:** Anil Sigdel, Anu Bista, Hardik Sapkota, Edwin van Teijlingen.

**Writing – review & editing:** Anil Sigdel, Anu Bista, Hardik Sapkota, Edwin van Teijlingen.

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
