## [Decision Letter · Decision Letter 0]

30 May 2022

PONE-D-22-05843Barriers in accessing family planning services in Nepal during the COVID-19 pandemic: A qualitative studyPLOS ONE

Dear Dr. Sigdel,

Thank you for submitting your manuscript to PLOS ONE. After careful consideration, we feel that it has merit but does not fully meet PLOS ONE’s publication criteria as it currently stands. Therefore, we invite you to submit a revised version of the manuscript that addresses the points raised during the review process.

 The authors should critically attend all the comments raised by the reviewers. Please mention the effect of COVID-19 pandemic on family planning services in Nepal. Please submit your revised manuscript by Jul 14 2022 11:59PM. If you will need more time than this to complete your revisions, please reply to this message or contact the journal office at plosone@plos.org. Please include the following items when submitting your revised manuscript:A rebuttal letter that responds to each point raised by the academic editor and reviewer(s). You should upload this letter as a separate file labeled 'Response to Reviewers'.A marked-up copy of your manuscript that highlights changes made to the original version. You should upload this as a separate file labeled 'Revised Manuscript with Track Changes'.An unmarked version of your revised paper without tracked changes. You should upload this as a separate file labeled 'Manuscript'.

We look forward to receiving your revised manuscript.

Kind regards,

Thae Maung Maung, MBBS, MSc (International Health), PhD

Academic Editor

PLOS ONE

Journal Requirements:

Reviewers' comments:

Reviewer's Responses to Questions

**Comments to the Author**

1. Is the manuscript technically sound, and do the data support the conclusions?

Reviewer #1: Yes

Reviewer #2: Partly

2. Has the statistical analysis been performed appropriately and rigorously? 

Reviewer #1: N/A

Reviewer #2: I Don't Know

3. Have the authors made all data underlying the findings in their manuscript fully available?

Reviewer #1: Yes

Reviewer #2: No

4. Is the manuscript presented in an intelligible fashion and written in standard English?

Reviewer #1: Yes

Reviewer #2: No

5. Review Comments to the Author

Reviewer #1: Thank you for letting me review this paper on barriers to family planning during the Covid19 pandemic. Overall, the study is well-conducted and address an important research question. However, I think the manuscript can be improved, mainly to have a clearer focus on the special circumstances of Covid19 and also when it comes to general recommendations that are not bound to the Covid19 situation. Both these concerns are addressed in the paper, but could be sharpened.

When I first read the abstract and the start of the Results section, I get the impression that this paper is about barriers to family planning in general. I think the relation to the lockdown and the pandemic can be made clearer throughout. Studies on general barriers to FP are plentifold and would not justify yet another study, but the special circumstances of a pandemic might. Especially if we can draw general conclusions on how to act in health emergencies.

In line 96, page 6, you state that “interviews explored the barriers to FP”. What about the special circumstances of the pandemic? How where they addressed? Please align with the aim of the study stated on lines 91-92.

This last point, that general insights can be drawn is important, since we are not likely to see an exact copy of the lock-down situation even if there will surely be more pandemics coming. On this note, I lack an overarching theme in the Results section. You have applied the socio-ecological framework, which is good, but what is the overarching lesson learnt from this study?

Line 117 has a broken reference

Line 111 mis-spelled PI

Line 115 Saturation was reached how?

I lack a section on reflexivity in the Discussion section.

I also lack methodological considerations. What about making telephone interviews, how may that have affected results?

Reviewer #2: Thank you for the opportunity to review this manuscript. Overall, it requires a significant edit to ensure meaning and content can be understood. This includes consistency and clarity with acronym use and a review of unclear phrasing (for eg, ‘havoc messages’, ‘unsecured feelings’ ‘people living of HIV’ etc.). Referencing also needs to be checked and updated throughout the document. Some references are incomplete or contain errors.

Below are some more specific questions and suggestions for consideration.

Introduction

• The WHO declared the COVID-19 outbreak a public health emergency of international concern on the 30th Jan 2020. A worldwide pandemic was declared March 11th 2020. Please correct this error.

• The cure rate and case fatality rate provided are not helpful data in the context of this study. More contextually informative data about the COVID-19 pandemic in Nepal is required. For example, number of confirmed cases, cases/million population, confirmed deaths, deaths/million population etc. A comment on the availability of COVID-19 testing or the reliability of national COVID-19 data in Nepal is also needed. If possible and available from the Nepal government, it would be helpful to also include COVID-19 data or context specifically from the 5/77 districts in Nepal that the authors purposely selected the 18 participants in their study.

• Reference 3 is not useful for readers to get an overall sense of the COVID-19 pandemic in Nepal, including at the time of data collection (Jan 15th – 25th 2021). You could consider referencing a global database such as:https://www.worldometers.info/coronavirus/country/nepal/

• A summary of the SRHR context in Nepal is needed here or in the study setting section below, with focus on access to contraception. For example, CPR, unmet need for contraception, contraceptive method-mix etc. Have previous studies looked into access barriers before the pandemic? If yes, what are they?

Materials and methods

• Please consider including data about phone ownership and access for women in Nepal.

• Did participants receive any incentives or reimbursements for their participation in the study?

• Page 6, line 97. Please define ‘regular clients’ within the context of this study.

• Page 7, line 99. Countries around the world implemented slightly different lockdowns as a way to manage COVID-19 spread. Please provide a brief description about what a ‘complete lockdown’ meant in the Nepal context, especially in regard to health care access more generally. Also, How long did the lockdown go for? This information could be included in the introduction.

• Please include information about the contraceptive use of participants. Were all participants accessing family planning services when they were approached to participate in the study? Were they new or continuing clients? What type of contraception was being used (for example. % LARC vs short-term)?

• If available, please include data on the number of children participants had.

Data collection:

• Page 8, table 1 – It is not clear how economic status or education level is determined and defined here. Please clarify.

• Please include information on who conducted the interviews.

• Some of the codes provided in Table 2 are not easily understood, especially in relation to sub-themes. Please edit codes for clarity. Alternatively, could the definition of several sample codes be included for readers to better understand the coding and analysis process?

• As a critical domain for qualitative inquiry, please include a brief section on research team reflexivity, including how interviewer characteristics could influence the participants response.

• Has there been any dissemination of research findings to participants, or member-checking?

Data analysis:

• Please include more detail about the transcription and translation process. What language were the transcriptions made in? Who conducted the transcription process? At what stage was translation conducted, by who, and how was it quality controlled to ensure accuracy and meaning was maintained, especially for direct quotes?

Results

• In general, more information and clarity is needed about what barriers existed before the pandemic. That is, were these barriers already present and were exacerbated or changed by the pandemic? Or did they not exist before?

• Please be specific when referring to COVID-19 infection or disease. For example, page 13 line 151.

Discussion:

• Additional references would strengthen this section and overall paper.

• Some repetition within the results section could also be removed.

• Page 26, line 431. Present study includes women of reproductive age 15 – 49 years. Should be 18 – 49 years?

6. PLOS authors have the option to publish the peer review history of their article (what does this mean?). If published, this will include your full peer review and any attached files.

Reviewer #1: No

Reviewer #2: No

---

## [Author Response · Author response to Decision Letter 0]

14 Aug 2022

Academic editor:

File naming was edited to comply with the style requirements. We hopefully have no

divergences from the style requirements now.

Data has been uploaded as supporting documents in the revised submission. 

Reviewer #1

Thank you for letting me review this paper on barriers to family planning during the Covid19 pandemic. Overall, the study is well-conducted and address an important research question. However, I think the manuscript can be improved, mainly to have a clearer focus on the special circumstances of Covid19 and also when it comes to general recommendations that are not bound to the Covid19 situation. Both these concerns are addressed in the paper, but could be sharpened.

Thank you for the suggestion. We updated the recommendations considering with COVID-19. Please refer the revised manuscript (clean version) discussion section (line #542-563) and conclusion section (line #565-591).

When I first read the abstract and the start of the Results section, I get the impression that this paper is about barriers to family planning in general. I think the relation to the lockdown and the pandemic can be made clearer throughout. Studies on general barriers to FP are plentifold and would not justify yet another study, but the special circumstances of a pandemic might. Especially if we can draw general conclusions on how to act in health emergencies.

We revise the conclusion section with more focus on how to act in health emergencies. Please refer the revised manuscript (cleaned version) conclusion section (line #565-591).

In line 96, page 6, you state that “interviews explored the barriers to FP”. What about the special circumstances of the pandemic? How where they addressed? Please align with the aim of the study stated on lines 91-92.

The sentences have been revised and few sentences were added that will highlights about the special circumstances of pandemic and the measures we ensured to address the gaps. Please refer the line #97-105 in revised manuscript (cleaned version). 

This last point, that general insights can be drawn is important, since we are not likely to see an exact copy of the lock-down situation even if there will surely be more pandemics coming. On this note, I lack an overarching theme in the Results section. You have applied the socio-ecological framework, which is good, but what is the overarching lesson learnt from this study?

Overarching lesson learnt section has been added in the result section in the revised manuscript. Please refer the line #426-437 in revised manuscript (cleaned version). 

Line 117 has a broken reference

Thank you for flagging. Corrected. 

Line 111 mis-spelled PI

Thank you for flagging. Corrected in the respective line. 

Line 115 Saturation was reached how

We noticed that the information was saturated because the last three interviews did not add any new information complementing to the objectives and scope of the study. This has been added in line #129-131 in revised manuscript (cleaned version).

I lack a section on reflexivity in the Discussion section

We added section on reflexivity on discussion part. Please refer the line #543-553 of revised manuscript (cleaned version).

I also lack methodological considerations. What about making telephone interviews, how may that have affected results?

Why we did telephone interview was explained in line # 99-102. We highlight some impact of telephone interview in line #554-564 (last paragraph of discussion section).

Reviewer #2

Thank you for the opportunity to review this manuscript. Overall, it requires a significant edit to ensure meaning and content can be understood. This includes consistency and clarity with acronym use and a review of unclear phrasing (for eg, ‘havoc messages’, ‘unsecured feelings’ ‘people living of HIV’ etc.). Referencing also needs to be checked and updated throughout the document. Some references are incomplete or contain errors.

Thank you for the suggestion. Use of acronyms has been made uniform throughout the manuscript and operation definition has been added of some of the phrasing. References have been checked and updated as suggested. 

Introduction

The WHO declared the COVID-19 outbreak a public health emergency of international concern on the 30th Jan 2020. A worldwide pandemic was declared March 11th 2020. Please correct this error.

Thank you for correction. Correction has been made. Please refer the line #52 in revised manuscript (Cleaned version). 

The cure rate and case fatality rate provided are not helpful data in the context of this study. More contextually informative data about the COVID-19 pandemic in Nepal is required. For example, number of confirmed cases, cases/million population, confirmed deaths, deaths/million population etc. A comment on the availability of COVID-19 testing or the reliability of national COVID-19 data in Nepal is also needed. If possible and available from the Nepal government, it would be helpful to also include COVID-19 data or context specifically from the 5/77 districts in Nepal that the authors purposely selected the 18 participants in their study.

The data as suggested has been added in the revised manuscript. Please refer the line #55-59 (in revised cleaned manuscript) which highlight the national level and study sites data from government source. 

Reference 3 is not useful for readers to get an overall sense of the COVID-19 pandemic in Nepal, including at the time of data collection (Jan 15th – 25th 2021). You could consider referencing a global database such as:https://www.worldometers.info/coronavirus/country/nepal/

Thank you for the suggestion. Worldometers references has been added and data updated from the source. 

A summary of the SRHR context in Nepal is needed here or in the study setting section below, with focus on access to contraception. For example, CPR, unmet need for contraception, contraceptive method-mix etc. Have previous studies looked into access barriers before the pandemic? If yes, what are they?

Thank you for the suggestion. This has been updated in study setting section. Please refer the line# 115-119 where different statistics in Nepal has been presented. 

Materials and methods

Please consider including data about phone ownership and access for women in Nepal.

This has been updated in study setting section. Please refer the line #115-119 on ownership of phone and female access to phone in Nepal in revised manuscript (cleaned version). 

Did participants receive any incentives or reimbursements for their participation in the study?

No. The participant did not receive any incentives or reimbursements for their participation in the study. Phone call was made by researcher in scheduled time and date so they need not need to pay for call.

Page 6, line 97. Please define ‘regular clients’ within the context of this study. 

Thank you. Definition has been added. Please refer page 7 line #103-104 in revised manuscript (cleaned version). 

Page 7, line 99. Countries around the world implemented slightly different lockdowns as a way to manage COVID-19 spread. Please provide a brief description about what a ‘complete lockdown’ meant in the Nepal context, especially in regard to health care access more generally. Also, How long did the lockdown go for? This information could be included in the introduction.

Thank you. The suggested information has been updated in lines #106-108 where we highlight about the duration of lockdown, definition of complete lockdown in study setting section. 

Please include information about the contraceptive use of participants. Were all participants accessing family planning services when they were approached to participate in the study? Were they new or continuing clients? What type of contraception was being used (for example. % LARC vs short-term)? If available, please include data on the number of children participants had.

Thank you for the suggestion. This has been updated in Table-1 (General Information about study participants)

Data Collection

Page 8, table 1 – It is not clear how economic status or education level is determined and defined here. Please clarify.

The definition has been updated in the footnote of Table-1. Education status has been determined according to Government of Nepal classification and economic status is self-reported by respondents. 

Please include information on who conducted the interviews.

Data was collected by female co-author. This has been updated in line #126 in revised manuscript. 

Some of the codes provided in Table 2 are not easily understood, especially in relation to sub-themes. Please edit codes for clarity. Alternatively, could the definition of several sample codes be included for readers to better understand the coding and analysis process?

Thank you for the suggestion. Codes has been revised and operation definition has been added. 

As a critical domain for qualitative inquiry, please include a brief section on research team reflexivity, including how interviewer characteristics could influence the participants response.

Team reflexivity has been added in discussion section. Please refer the line # 543-553 in revised manuscript (cleaned Version). 

Has there been any dissemination of research findings to participants, or member-checking?

Yes. We checked with health service providers after completion of interviews in each district. This has been updated in line #133-135 in revised manuscript (cleaned version). 

Data analysis

Please include more detail about the transcription and translation process. What language were the transcriptions made in? Who conducted the transcription process? At what stage was translation conducted, by who, and how was it quality controlled to ensure accuracy and meaning was maintained, especially for direct quotes?

This has been updated and revised in line #142-144 of revised manuscript (cleaned version). Data transcription was in Nepali by female co-author which was checked by PI and another author to ensure the quality. 

Results

In general, more information and clarity is needed about what barriers existed before the pandemic. That is, were these barriers already present and were exacerbated or changed by the pandemic? Or did they not exist before?

Thank you for the suggestion. Some of these barriers exist before pandemic but the severity has been more during pandemic, which we have highlighted in overarching result in result section. Please refer the line #425-436 of revised manuscript (cleaned version). 

Please be specific when referring to COVID-19 infection or disease. For example, page 13 line 151.

Thank you. Revised as Infection throughout the report. 

Discussion:

Additional references would strengthen this section and overall paper.

Six additional references have been added to strengthen the paper. 

Some repetition within the results section could also be removed.

Revised and some references has been removed. 

Page 26, line 431. Present study includes women of reproductive age 15 – 49 years. Should be 18 – 49 years?

Thank You. Correction has been made. Please refer the line #464 in revised manuscript.

---

## [Decision Letter · Decision Letter 1]

2 Jan 2023

PONE-D-22-05843R1Barriers in accessing family planning services in Nepal during the COVID-19 pandemic: A qualitative studyPLOS ONE

Dear Dr. Sigdel,

Thank you for submitting your manuscript to PLOS ONE. After careful consideration, we feel that it has merit but does not fully meet PLOS ONE’s publication criteria as it currently stands. Therefore, we invite you to submit a revised version of the manuscript that addresses the points raised during the review process.

The manuscript is interesting and informative but it still needs some improvement. Please provide the specific discussion points based on the your study's results.

We look forward to receiving your revised manuscript.

Kind regards,

Thae Maung Maung, MBBS, MSc (International Health), PhD

Academic Editor

PLOS ONE

Reviewers' comments:

Reviewer's Responses to Questions

**Comments to the Author**

1. If the authors have adequately addressed your comments raised in a previous round of review and you feel that this manuscript is now acceptable for publication, you may indicate that here to bypass the “Comments to the Author” section, enter your conflict of interest statement in the “Confidential to Editor” section, and submit your "Accept" recommendation.

Reviewer #3: (No Response)

Reviewer #4: All comments have been addressed

2. Is the manuscript technically sound, and do the data support the conclusions?

Reviewer #3: Yes

Reviewer #4: Yes

3. Has the statistical analysis been performed appropriately and rigorously? 

Reviewer #3: Yes

Reviewer #4: N/A

4. Have the authors made all data underlying the findings in their manuscript fully available?

Reviewer #3: Yes

Reviewer #4: Yes

5. Is the manuscript presented in an intelligible fashion and written in standard English?

Reviewer #3: Yes

Reviewer #4: No

6. Review Comments to the Author

Reviewer #3: Thank you for allowing me to review the manuscript entitled " Barriers in accessing family planning services in Nepal during the COVID-19 pandemic: A qualitative study". It is indeed an important topic as majority of the routine essential services like family planning have been interrupted by the COVID 19 pandemic. In general, the study was well structured, and the objectives were clear. However, the result session seems to sway to the general barriers on family planning than the time of the COVID 19. In many of the previous studies we do know that individual factors like education, socioeconomic status, rural, urban differences and poor knowledge: family factor like spousal consent and role of mother in laws and elders: institutional factors like physical infrastructures and behaviour of health staff all contribute to barriers in accessing family planning. But the focus of the study is not general, thus it should emphasize more on the COVID related factors. Having said that there are good results such as misinformation about COVID 19, travel restrictions barriers and limited choice of family planning methods. Discussion and the recommendations also seem very general, and it does not focus much on the COVID 19 period and comparison articles are very general and sometimes it does not fit the contextual factors of the study and the situation. I think this can be improved by searching for studies related to COVID 19 period in similar settings or other shocks like earthquakes in Nepal.

Abstract

As the study aim is to " explore barriers faced by women while seeking family planning services in Nepal during COVID-19 lockdown" line 92/93, the abstract should highlight more of the COVID 19 related findings and conclusions to bring light to the objective of the study.

Introduction

Global COVID 19 pandemics have been addressed and the situation of the Nepal context and the lock down periods need to be spell out clearly with more emphasis on case load, fatality and vaccine coverage and most effected districts.

If there are previous studies looking into barriers in accessing family planning service in general, more information should be added as the focus is on family planning. Please also add the interruption of routine service data during the COVID 19.

Methodology

For the qualitative methodology reporting, the COREQ (Consolidated Criteria for Reporting Qualitative Studies) is a gold atandard to follow and in this study there are some methodological issues that need to be addressed.

1. Research team composition and who were they and who did the interviews. Any bias in selection of particpants.

2. Line 111-112 the sample of the women in the study were selected from " three geographical areas (Mountain, Hills and Terai) of Nepal. These clinics are the outlet of one NGO working on SRH whose clinics are accredited by Family Welfare Division of Nepal and has stayed open during the lockdown". But in the study findings there were no differences mentioned between the geographical areas, please explain? Also is there a reason why you only chose the 5 district ( More covid cases??)

3. You have chosen only women from outlet of one NGO on SRH clinic, and it is hard to draw conclusion to the whole Nepal without identifying the bias in qualitative sampling as well.

4. Were there any difficulties encountered during the telephone interviews? 30-40 min long talks on the phone and compensation for the phone charges needed to be spell out clearly. Please also mentioned reflexibility of the phone interviews within the team.

5. Please also reconsider the coding frame for more clarity of the themes and sub themes and the code definitions.

6. In line 115 saturation of the findings was reached but please mention how.

Results

Results session need to be readjusted to pay more attention to the COVID 19 related barriers rather than writing it in general.

Discussion

Please do not repeat the results again in the discussing session.

Line 449-453: You talked about the low education and poverty for your result but in the comparison, it was with urban and rural. Please check and review.

Your study includes women of reproductive age 19-49 and you often compare with the study of adolescent 15-24 which is hard to understand. Please find a suitable reference to compare the factors.

Line 469-473: You have mentioned about the decision-making power of women with better education and living in nuclear family. However, you related to the mother-in-law as a decision maker in the comparison study which is confusing. Please aligned your stand and rephrase.

Line 486-495: I do understand that misinformation on FP may hinder women to seek services but how does that relate to COVID 19 is not clear. Does COVID 19 lockdown may have an indirect effect on information seeking for FP.

Reviewer #4: This study informs me the barriers to access FP services during the Covid-19 in Nepal. It is good and informative to readers but the following points should be consider to improve this study before publications.

Introduction

It should be more elaborative to justify why this study should be conducted in Nepal. While the situation related with impacts of Covid-19 on FP services in general are well elaborative, the situation of Nepal is not described in introduction although some sentences are described in study settings. It will be more attractive to reader if the situation of Nepal would be mentioned before the study aim as why thus study should be done inevitably.

Methods

The reference for socio-ecological model should be added in line 145.

Results

According to the study aim, this study will identify the barriers. At the individual level, some of the quotes include both the barriers and facilitators or how to overcome the barriers to receive FP during the Covid-19. For examples, low self-confidence, inadequate knowledge, perceived importance. If so, it could be confused to readers even though these are informative. But, in the rest of the quotes, there are only barriers. I think it should be consistent with the objective and the whole structure.

Under the overarching lesson learnt, there are some sentences related with impact of Covid-19 on FP for global condition. I understand that it should be study specific overall acquired knowledge after conducting study. It should be considered again to be more relevant. Current sentences may be useful to be in discussion part.

Discussion

At line 502, it is not clear for “commodity based contraception”. Is it the same as modern contraception?

Also it was discussed about elaborately for fear of side effects while it was not mentioned in results or conclusion as important barriers.

In discussion, most the references used to compare were conducted before Covid-19. It would be more informative if there are compared with the study or literatures for specific Covid-19 context. And, some the barriers are due to Covid-19 and have not existed before Covid-19. Some are existing barriers for FP but more exaggerated due to Covid-19. This concept should be discussed thoroughly to be clear for readers.

In the discussion, I would like to suggest considering again to be more attractive. Now, the comparative findings are written for each barriers and then there no overall conceptualization of this study to convince to readers. It should also be connected with overall lesson learnt from results. Now it is difficult to follow what the authors want to highlight the key discussion points. Then, it should be connected with conclusion. The qualitative discussion should be like a good narrative storytelling to be more attractive to readers.

Strengths and limitations should be sub-tiled to clear for readers.

7. PLOS authors have the option to publish the peer review history of their article (what does this mean?). If published, this will include your full peer review and any attached files.

Reviewer #3: **Yes: **Kyu Kyu Than

Reviewer #4: **Yes: **Khaing Nwe Tin

---

## [Author Response · Author response to Decision Letter 1]

15 Mar 2023

We thank the academic editor and the two reviewers for their comments on our manuscript. Below is our response to each point raised by the reviewers. We hope that we satisfyingly addressed them and that the manuscript will be now suited for publication. 

Sincerely,

On behalf of all authors,

Anil Sigdel

Reviewer #3

Abstract

As the study aim is to " explore barriers faced by women while seeking family planning services in Nepal during COVID-19 lockdown" line 92/93, the abstract should highlight more of the COVID 19 related findings and conclusions to bring light to the objective of the study.

Thank you for the suggestions. Key highlights related to COVID-19 have been added in the abstract. Please review the revised abstract section. 

Introduction

Global COVID 19 pandemics have been addressed and the situation of the Nepal context and the lock down periods need to be spell out clearly with more emphasis on case load, fatality and vaccine coverage and most effected districts.

If there are previous studies looking into barriers in accessing family planning service in general, more information should be added as the focus is on family planning. Please also add the interruption of routine service data during the COVID 19.

The information has been updated in lines 60-66 and some study context has also been provided in study setting (line 126-140) in revised manuscript.

Methodology

For the qualitative methodology reporting, the COREQ (Consolidated Criteria for Reporting Qualitative Studies) is a gold atandard to follow and in this study there are some methodological issues that need to be addressed.

1. Research team composition and who were they and who did the interviews. Any bias in selection of particpants.

Information related to research team composition and one conducting interview has been updated in line 142-144. Further, regarding bias has been updated in line 120-124 in revised manuscript. 

2. Line 111-112 the sample of the women in the study were selected from " three geographical areas (Mountain, Hills and Terai) of Nepal. These clinics are the outlet of one NGO working on SRH whose clinics are accredited by Family Welfare Division of Nepal and has stayed open during the lockdown". But in the study findings there were no differences mentioned between the geographical areas, please explain? Also is there a reason why you only chose the 5 district ( More covid cases??)

Yes. The Prime reason is high number of COVID-19 cases in those five districts i.e. 39% of total COVID-19 cases Regarding outlet and representation, this NGO has countrywide clinic outlets and clinics in districts with high COVID-19 cases in each ecological region has been selected. 

3. You have chosen only women from outlet of one NGO on SRH clinic, and it is hard to draw conclusion to the whole Nepal without identifying the bias in qualitative sampling as well.

Thank you. The bias in qualitative sampling has been highlighted in line 120-124 in revised manuscript.

4. Were there any difficulties encountered during the telephone interviews? 30-40 min long talks on the phone and compensation for the phone charges needed to be spell out clearly. Please also mentioned reflexibility of the phone interviews within the team.

Reflexibility on phone interview has been reflected in line 150-160 in revised manuscript. Please refer the revised manuscript. 

5. Please also reconsider the coding frame for more clarity of the themes and sub themes and the code definitions.

Thank you. This has been considered. Please refer the updated code tree in revised manuscript.

6. In line 115 saturation of the findings was reached but please mention how.

This has been reflected in line 161-162. We noticed that the information was saturated because the last three interviews did not add any new information complementing the objectives and scope of the study.

Results

Results session need to be readjusted to pay more attention to the COVID 19 related barriers rather than writing it in general.

Thank you. This has been readjusted. Please refer to the revised manuscript. 

Discussion

Please do not repeat the results again in the discussing session.

Line 449-453: You talked about the low education and poverty for your result but in the comparison, it was with urban and rural. Please check and review.

Thank you. Comparisons with other relevant literatures have been made in line 513-523 in revised manuscript. 

Your study includes women of reproductive age 19-49 and you often compare with the study of adolescent 15-24 which is hard to understand. Please find a suitable reference to compare the factors.

Relevant literature has been updated. Please refer the line 512-523 in revised manuscript.

Line 469-473: You have mentioned about the decision-making power of women with better education and living in nuclear family. However, you related to the mother-in-law as a decision maker in the comparison study which is confusing. Please aligned your stand and rephrase.

This has been revised and compared with relevant literatures. Please refer to line 535-540 in revised manuscript.

Line 486-495: I do understand that misinformation on FP may hinder women to seek services but how does that relate to COVID 19 is not clear. Does COVID 19 lockdown may have an indirect effect on information seeking for FP.

Yes. The COVID-19 lockdown have indirect effect on information for FP as most of the information were confined with COVID-19 and people do not have access with right information to address the misconception regarding FP, which affect in up-take of FP services. 

Reviewer #4: 

Introduction

It should be more elaborative to justify why this study should be conducted in Nepal. While the situation related with impacts of Covid-19 on FP services in general are well elaborative, the situation of Nepal is not described in introduction although some sentences are described in study settings. It will be more attractive to reader if the situation of Nepal would be mentioned before the study aim as why thus study should be done inevitably.

Thank You for the suggestion. Additional information on situation of Nepal has been updated in introduction section in revised manuscript. Please refer the line 101-107.

Methods

The reference for socio-ecological model should be added in line 145.

Thank You. Updated. 

Results

According to the study aim, this study will identify the barriers. At the individual level, some of the quotes include both the barriers and facilitators or how to overcome the barriers to receive FP during the Covid-19. For examples, low self-confidence, inadequate knowledge, perceived importance. If so, it could be confused to readers even though these are informative. But, in the rest of the quotes, there are only barriers. I think it should be consistent with the objective and the whole structure.

Thank you for the suggestion. This has been corrected. 

Under the overarching lesson learnt, there are some sentences related with impact of Covid-19 on FP for global condition. I understand that it should be study specific overall acquired knowledge after conducting study. It should be considered again to be more relevant. Current sentences may be useful to be in discussion part.

Thank you for the suggestion. This has been considered in revised manuscript.

Discussion

At line 502, it is not clear for “commodity based contraception”. Is it the same as modern contraception?

Yes. This is a modern contraception. Community based contraception has been replaced with modern contraception in revised manuscript. 

Also it was discussed about elaborately for fear of side effects while it was not mentioned in results or conclusion as important barriers.

This has been updated in result part mostly in social stigma section. Thank You

In discussion, most the references used to compare were conducted before Covid-19. It would be more informative if there are compared with the study or literatures for specific Covid-19 context. And, some the barriers are due to Covid-19 and have not existed before Covid-19. Some are existing barriers for FP but more exaggerated due to Covid-19. This concept should be discussed thoroughly to be clear for readers.

Thank you. Findings were compared with several studies conducted during COVID-19 and Ebola Crisis in revised discussion section. Further, different sections have been revised to show how COVID-19 further exaggerated the pre-existing barriers. 

In the discussion, I would like to suggest considering again to be more attractive. Now, the comparative findings are written for each barriers and then there no overall conceptualization of this study to convince to readers. It should also be connected with overall lesson learnt from results. Now it is difficult to follow what the authors want to highlight the key discussion points. Then, it should be connected with conclusion. The qualitative discussion should be like a good narrative storytelling to be more attractive to readers.

Thank you. Key findings and highlights are discussed in discussion sections as suggested. Please refer to the revised discussion section of revised manuscript. 

Strengths and limitations should be sub-tiled to clear for readers.

Thank You. This has been updated.

---

## [Decision Letter · Decision Letter 2]

10 Apr 2023

PONE-D-22-05843R2Barriers in accessing family planning services in Nepal during the COVID-19 pandemic: A qualitative studyPLOS ONE

Dear Dr. Sigdel,

Thank you for submitting your manuscript to PLOS ONE. After careful consideration, we feel that it has merit but does not fully meet PLOS ONE’s publication criteria as it currently stands. Therefore, we invite you to submit a revised version of the manuscript that addresses the points raised during the review process.

Dear Author, The manuscript looks much improved than the previous version and one step near to publication.However, there is one minor comment from one reviewer. Please take careful consideration and revise accordingly.==============================

We look forward to receiving your revised manuscript.

Kind regards,

Thae Maung Maung, MBBS, MSc (International Health), PhD

Academic Editor

PLOS ONE

Journal Requirements:

Reviewers' comments:

Reviewer's Responses to Questions

**Comments to the Author**

1. If the authors have adequately addressed your comments raised in a previous round of review and you feel that this manuscript is now acceptable for publication, you may indicate that here to bypass the “Comments to the Author” section, enter your conflict of interest statement in the “Confidential to Editor” section, and submit your "Accept" recommendation.

Reviewer #3: All comments have been addressed

Reviewer #4: (No Response)

2. Is the manuscript technically sound, and do the data support the conclusions?

Reviewer #3: Yes

Reviewer #4: Yes

3. Has the statistical analysis been performed appropriately and rigorously? 

Reviewer #3: N/A

Reviewer #4: N/A

4. Have the authors made all data underlying the findings in their manuscript fully available?

Reviewer #3: Yes

Reviewer #4: Yes

5. Is the manuscript presented in an intelligible fashion and written in standard English?

Reviewer #3: Yes

Reviewer #4: Yes

6. Review Comments to the Author

Reviewer #3: The authors have addressed all the comments given to them and they have revised their manuscipts well.

Reviewer #4: Thank you for sharing the revised manuscripts which improves a lot and has been revised most of the comment. But one comment related with Overarching lesson learnt; "Under the overarching lesson learnt, there are some sentences related with impact of Covid-19 on FP for global condition. I understand that it should be study specific overall

acquired knowledge after conducting study. It should be considered again to be more relevant. Current sentences may be useful to be in discussion part", was not revised yet.

Actually, in the results part, the findings should be from your study/ information that you collected during interviews. It should not be from other literature/ reference. These references will be useful in discussion part. Here, it should be overall findings of your study should be conceptualized to convince to the readers.

7. PLOS authors have the option to publish the peer review history of their article (what does this mean?). If published, this will include your full peer review and any attached files.

Reviewer #3: **Yes: **Kyu Kyu Than

Reviewer #4: No

---

## [Author Response · Author response to Decision Letter 2]

17 Apr 2023

Thank you for the suggestion. This has been updated in the revised manuscript as, "The study found that along with the prevailing barriers in the uptake of FP, the global pandemic like COVID-19 have significant role in introducing/adding new additional barriers in uptake of FP services. The emergence of COVID-19 leads to movement restriction, financial hardship due to loss of job, un-acceptance of FP as essential health services, increased time at home with husband or parents, violation of privacy, increased waiting time at health facilities, limited outreach services by community health workers, delayed prepositioning of FP commodity and diversion of health workers to respond COVID-19 were noted as the additional barriers in uptake of FP services by women in low-and-middle-income countries like Nepal. This clearly implies that public health efforts, particularly SRH programs, should consider the additional challenges that occur as a result of the pandemic during the design and execution of public health programs." Please refer the line number 483-493. Thank You

---

## [Decision Letter · Decision Letter 3]

19 Apr 2023

Barriers in accessing family planning services in Nepal during the COVID-19 pandemic: A qualitative study

PONE-D-22-05843R3

Dear Dr. Sigdel,

We’re pleased to inform you that your manuscript has been judged scientifically suitable for publication and will be formally accepted for publication once it meets all outstanding technical requirements.

Kind regards,

Thae Maung Maung, MBBS, MSc (International Health), PhD

Academic Editor

PLOS ONE

Additional Editor Comments (optional):

Reviewers' comments:

Reviewer's Responses to Questions

**Comments to the Author**

1. If the authors have adequately addressed your comments raised in a previous round of review and you feel that this manuscript is now acceptable for publication, you may indicate that here to bypass the “Comments to the Author” section, enter your conflict of interest statement in the “Confidential to Editor” section, and submit your "Accept" recommendation.

Reviewer #4: All comments have been addressed

2. Is the manuscript technically sound, and do the data support the conclusions?

Reviewer #4: Yes

3. Has the statistical analysis been performed appropriately and rigorously? 

Reviewer #4: N/A

4. Have the authors made all data underlying the findings in their manuscript fully available?

Reviewer #4: Yes

5. Is the manuscript presented in an intelligible fashion and written in standard English?

Reviewer #4: Yes

6. Review Comments to the Author

Reviewer #4: Thank you for sending the revised version. Now I hope it can be published after adjusting the format from the journal.

7. PLOS authors have the option to publish the peer review history of their article (what does this mean?). If published, this will include your full peer review and any attached files.

Reviewer #4: **Yes: **Khaing Nwe Tin

---

## [Editor Report · Acceptance letter]

27 Apr 2023

PONE-D-22-05843R3 

Barriers in accessing family planning services in Nepal during the COVID-19 pandemic: A qualitative study 

Dear Dr. Sigdel:

I'm pleased to inform you that your manuscript has been deemed suitable for publication in PLOS ONE. Congratulations! Your manuscript is now with our production department. 

Kind regards, 

on behalf of

Dr. Thae Maung Maung 

Academic Editor

PLOS ONE